# The Potential of *Aloe vera* and *Opuntia ficus-indica* Extracts as Biobased Agents for the Conservation of Cultural Heritage Metals

**DOI:** 10.3390/metabo15060386

**Published:** 2025-06-10

**Authors:** Çağdaş Özdemir, Lucia Emanuele, Marta Kotlar, Marina Brailo Šćepanović, Laura Scrano, Sabino Aurelio Bufo

**Affiliations:** 1Department of Basic and Applied Sciences, Basilicata University, 85100 Potenza, Italy; cagdas.ozdemir@unibas.it; 2Department of Art and Restoration, University of Dubrovnik, 20000 Dubrovnik, Croatia; marta.kotlar@unidu.hr; 3Department of Applied Ecology, University of Dubrovnik, 20000 Dubrovnik, Croatia; marina.brailo@unidu.hr; 4Department for Humanistic, Scientific and Social Innovation, Basilicata University, 85100 Potenza, Italy; laura.scrano@unibas.it; 5CNR-IRSA Istituto di Ricerca Sulle Acque, Via Roma, 3, 74123 Taranto, Italy; 6Department of Geography, Environmental Management and Energy Studies, University of Johannesburg, 2000 Johannesburg, South Africa

**Keywords:** *Aloe vera* (L.) Burm.f., antibacterial activity, biocorrosion, biofilm inhibition, copper alloys, cultural heritage conservation, natural corrosion inhibitors, *Opuntia ficus-indica* (L.) mill., plant secondary metabolites, *Pseudomonas aeruginosa*

## Abstract

Background/Objectives: Biocorrosion, driven by microbial colonization and biofilm formation, poses a significant threat to the integrity of metal artifacts, particularly those composed of copper and its alloys. *Pseudomonas aeruginosa*, a bacterial species that reduces nitrates, plays a key role in this process. This study explores the potential of two metabolite-rich plant extracts, *Aloe vera* and *Opuntia ficus-indica*, as sustainable biobased inhibitors of microbial-induced corrosion (MICOR). Methods: The antibacterial and antibiofilm activities of the extracts were evaluated using minimal inhibitory concentration (MIC) assays, time-kill kinetics, and biofilm prevention and removal tests on copper, bronze, and brass samples. Spectrophotometric and microbiological methods were used to quantify bacterial growth and biofilm density. Results: Both extracts exhibited significant antibacterial activity, with MIC values of 8.3% (*v*/*v*). *A. vera* demonstrated superior bactericidal effects, achieving reductions of ≥3 log_10_ in bacterial counts at lower concentrations. In antibiofilm assays, both extracts effectively prevented biofilm formation and reduced established biofilms, with *A. vera* exhibiting greater efficacy against them. The active metabolites—anthraquinones, phenolics, flavonoids, and tannins—likely contribute to these effects. Conclusions: These findings highlight the dual role of *A. vera* and *O. ficus-indica* extracts as both corrosion and biocorrosion inhibitors. The secondary metabolite profiles of these plants support their application as eco-friendly alternatives in the conservation of metal cultural heritage objects.

## 1. Introduction

Microorganisms, including bacteria, fungi, and algae, play a significant role in the corrosion of metal artifacts, particularly those exposed to terrestrial or aquatic environments. The extent and nature of corrosion depend on the type of metal and environmental conditions, but microbial-induced corrosion (MICOR) is often underestimated in the context of cultural heritage conservation, despite its well-documented occurrence in natural settings [1].

Among the microbial agents, bacteria—especially nitrate-reducing bacteria (NRB)—are frequently implicated in MICOR. NRBs can constitute up to 50% of the microbial population in aquatic systems [2,3], with *Pseudomonas aeruginosa* being one of the most studied species due to its aggressive biofilm-forming capacity and its role in accelerating corrosion on various metals, including copper [4,5,6]. Given its prevalence in the Mediterranean Sea [7] and the regional relevance of the authors’ institutions, *P. aeruginosa* was selected as the target organism in this study.

Copper and its alloys (bronze and brass) are historically significant materials in Mediterranean cultural heritage. Despite copper’s occasional use as an antimicrobial agent [7], it remains susceptible to MICOR, particularly in marine environments [8]. Many artifacts recovered from underwater archaeological sites are composed of these metals, making their preservation a priority.

In recent years, plant-based corrosion inhibitors have gained attention as sustainable alternatives to synthetic chemicals. Extracts from *Aloe vera* and *Opuntia ficus-indica* (prickly pear), both native to the Mediterranean region, are rich in secondary metabolites with known antioxidant and antimicrobial properties [9,10,11,12,13,14]. *A. vera* contains anthraquinones (e.g., aloin, emodin), phenolic compounds, saponins, and carotenoids, which contribute to its antibacterial and antioxidant activities [15,16,17,18]. Similarly, *O. ficus-indica* is rich in phenolics, flavonoids, and tannins, which have demonstrated corrosion-inhibiting and antimicrobial effects [19].

Although these plants have been studied as corrosion inhibitors for modern and archaeological metals, their potential to inhibit biocorrosion, specifically by targeting microbial biofilms, remains underexplored. Furthermore, conservation practices require that any treatment be minimally invasive and reversible, making natural, biodegradable substances particularly attractive.

The aim of this study is to evaluate the antibacterial and antibiofilm activities of *A. vera* and *O. ficus-indica* extracts against *P. aeruginosa* and to assess their effectiveness in preventing and removing biofilms on copper, bronze, and brass surfaces. By integrating microbiological assays with conservation science, this research aims to identify eco-friendly, reversible strategies for preserving metal cultural heritage artifacts.

## 2. Methods

This study used a bacterial species, two plant extracts, various metal samples, and an antibiotic. Two types of experiments were performed: antibacterial and antibiofilm. Depending on the specific test, multiple samples were prepared.

### 2.1. P. aeruginosa

*P. aeruginosa* is a widespread gram-negative bacterial species. We used a culture obtained from the Public Health Institute of Dubrovnik-Neretva County for our target. For the liquid cultures (in biological tubes), tryptic soy broth (TSB) (Komed, Brezje, Croatia) was used for bacterial growth, with a concentration of 10^5^ CFU (colony-forming units) per mL. The count was done with Thoma slides under a light microscope. For the solid cultures (in Petri dishes), a tryptic glucose–yeast agar culture medium (Biolife, Milan, Italy) was used for bacterial growth. Pure bacterial cultures were isolated after incubating the tubes or dishes at 37 °C for 24 h.

### 2.2. The Plant Extracts

Three healthy and fresh cladodes, approximately 20 to 25 cm in length for *O. ficus-indica* and 15 to 20 cm leaves for *A. vera*, were selected from plants collected in late spring from ornamental specimens growing in a private garden in the city of Dubrovnik, Croatia.

The extract of *O. ficus-indica* was prepared using the maceration method [20]. The cladodes were cleaned with ethanol and peeled. The inner pulp was diced, placed in a beaker, and covered with distilled water at a 20% m/v ratio. After 24 h in the dark, the gel was formed and collected by filtration. A part of the gel was used to measure the concentration of the extract by complete evaporation of water in a domestic oven at 70–75 °C [20]. This process required several hours and yielded a concentration of 7 g of dried gel per kg of raw material (0.7% *w*/*w*). For the tests reported below, the non-dried gel was used because, thanks to its higher viscosity, it was easier to spread the gel on metal surfaces. The dried gel consists of a fibrous material composed of polysaccharides and protein residues [21] and contains a significant amount of secondary metabolites, characterized according to Mokrani et al. and references cited therein [22].

The *A. vera* extraction procedure consisted of wiping the leaf with ethanol, peeling it, and cutting it horizontally into two parts along its entire length. The internal gel was collected using a spoon and blended to achieve a uniform consistency without the addition of water, as it already consisted of 99.5% water (*w*/*w*). The remaining 0.5% (*w*/*w*) of the solid material comprises various compounds, including several secondary metabolites, which were characterized using literature methods [23,24,25,26].

### 2.3. Metal Samples

Several metal samples were prepared for testing purposes. The samples (3 cm^2^) were cut from pure copper, bronze (a mixture of copper and tin), and brass (a mixture of copper and zinc) sheets with a thickness of 1 mm. The samples were cleaned of impurities and patina with sandpaper, then sterilized with 96% ethanol and autoclaved (Nüve OT 012). These techniques ensured the metal surface was clean, validating further test results.

### 2.4. Antibiotic

The common antibiotic azithromycin (Sumamed^®^) was used during the experiments. The concentration of azithromycin was 200 mg/5 mL, as indicated in the leaflet provided by the manufacturer in the package. The antibiotic was used as a negative control to compare its antibacterial efficiency with the potential bactericidal activity of the plant extracts.

### 2.5. Antibacterial Experiments

The antibacterial activity of plant extracts was assessed by determining the minimal inhibitory concentration (MIC) and by performing a time-kill test to evaluate the extracts’ concentration-dependent and time-dependent effectiveness in eliminating the *P. aeruginosa.* For this purpose, gel extracts of *A. vera* and *O. ficus-indica* were separately dispersed in distilled water at four different concentrations, 12.5%, 25%, 50%, and 100% (***v*****/*****v***), ensuring a total volume of 1 mL for each mixture.

Eight microbiological test tubes were filled with 5 mL of TSB and 25 μL of *P. aeruginosa* (10^5^ CFU/mL). We ensured the exact concentration using a Thoma slide and a light microscope. The bacterial sample taken from the tube was placed on the Thoma slide, and the number of bacteria on the squares was counted, resulting in a calculated count of 10^8^ CFU/mL. Then we performed 3-fold serial dilutions with tryptic soy broth (TSB) and checked the density again using a Thoma slide under a light microscope.

Then, 1 mL of each *O. ficus-indica* extract suspension was added to one of four tubes, and 1 mL of each *A. vera* extract suspension to the other four. Based on the proportion of the plant extracts in prepared mixtures, the volume concentrations of plant extracts in the MIC tubes were calculated to be 2.075% (***v*****/*****v***), 4.15% (***v*****/*****v***), 8.3% (***v*****/*****v***), and 16.6% (***v*****/*****v***), respectively. The samples containing TSB, *P. aeruginosa*, and the *O. ficus-indica* extract suspensions were labeled as 1–4, while those containing the *A. vera* extract suspensions were labeled as 5–8.

In addition, two control samples were prepared: one tube containing only 5 mL of TSB was used as the negative control (labeled as NC), while another tube containing 5 mL of TSB and 25 μL of bacterial suspension served as the positive control (labeled as PC). All ten samples (NC, PC, 1, 2, 3, 4, 5, 6, 7, and 8) were incubated at 37 °C for 24 h. During incubation, subsamples were taken at 3, 6, 18, and 24 h to assess the time-dependent effectiveness of the extracts in eliminating *P. aeruginosa* by spectrophotometric analysis and the spread plate method [27,28].

MIC defines in vitro the resistance levels of specific bacteria to a given antibiotic. The MIC value is the lowest concentration at which bacterial growth is completely inhibited. It can be determined by observing and comparing the turbidity of solutions: the dilution with the lowest concentration of the test material that exhibits no turbidity is the MIC [28]. In this study, we determined the MIC of the antibiotic and the two plant extracts.

For the spectrophotometric analysis, 2 μL of each subsample was used to perform absorbance measurements at 625 nm by the spectrophotometer IMPLEN-NanoPhotometer N50 version (Munich, Germany) [29,30]. The absorptions obtained were used to quantify the number of bacteria in the tube using the equation ***A*** = **εc*ℓ***, where A is absorption, ε is the extinction coefficient, c is the concentration of the bacterial solution, and ℓ is the path length.

For the spread-plate method (a microbiological technique used to grow and count bacteria), 1 mL of each subsample was taken and then diluted three times by a factor of ten. Specifically, 1 mL of the original solution was added to 9 mL of distilled water to obtain the first 1:10 dilution (10^−1^). This process was repeated twice to achieve the second and third dilutions (10^−2^ and 10^−3^, respectively). The result is four solutions with different concentrations.

From each dilution, 100 μL was taken and spread evenly over the surface of a solid nutrient agar plate, allowing bacteria to grow into visible colonies at 37 °C for 24 h. The bacterial colonies grown on plates were manually counted [27], and the number of colonies was estimated by the equation CFU/mL = (No. of colonies × dilution factor)/culture volume.

All experiments were replicated three times.

### 2.6. Antibiofilm Experiments

Only after completing the antibacterial experiments was it possible to evaluate the effect of the same plant extracts on the prepared metal samples. Two aspects of the extracts’ anti-biofilm properties were investigated: their ability to prevent bacterial biofilm formation on the metal surface and their effectiveness in treating an already established biofilm.

To test the ability of extracts to prevent bacterial biofilm formation, four solutions were prepared and applied to metal samples, including copper, bronze, and brass, resulting in a total of 16 samples. The first solution consisted of 500 μL of bacterial suspension of *P. aeruginosa* and served as a positive control. The other three solutions were prepared by mixing 500 μL of bacterial suspension with 500 μL of one of the antibacterial agents: the antibiotic (azithromycin) as a negative control, the extract of *O. ficus-indica*, and the extract of *A. vera*. The metal samples treated with these solutions were incubated at 37 °C for 48 h. After these two days, 100 μL of distilled water was placed on the metal surface and drawn back into the pipette (pipetting). After the third pipetting, the liquid retrieved from the metal surface into the pipette was transferred to tryptic glucose yeast agar plates. The transferred 100 μL of liquid was spread evenly across the plate using a completely sterile spreader. After 48 h of incubation at 37 °C, the density of the colonies in the Petri dishes was examined. The 12 samples obtained were labeled as shown in Table 1. After 48 h of incubation at 37 °C, the colonies on the plates were compared for differences in bacterial density.

To test the effectiveness of extracts in treating an already established biofilm, three samples of each metal were first exposed to 500 μL of bacterial suspension. The samples were then incubated at 37 °C for 72 h to allow bacterial growth and biofilm formation on the metal surfaces. After incubation, one sample of each metal was left untreated as a control, while the remaining two samples were each treated with 500 μL of *O. ficus indica* extract or *A. vera* extract and then incubated for an additional 48 h. Then 100 μL of distilled water was put onto the metal surface (as described above). The liquid was retrieved back into the pipette and then transferred to a microscope slide for staining. Then it was stained with a methylene blue solution prepared by adding 0.8 g of methylene blue powder to 80 mL of distilled water and mixing for 45 min at 600 rpm (revolutions per minute) on a magnetic stirrer. In this way, when the slide was placed under the microscope (Olympus optical, Tokyo, Japan, CH20BIMF200), it was possible to see the stained blue bacteria against a colorless background and visualize the differences in bacterial density.

### 2.7. Evaluation and Cleaning of Metal Samples

All samples were visually inspected after biofilm removal tests to verify the absence of residues or surface changes. The samples were then cleaned with a damp cotton swab to remove traces of extract from the surface.

After cleaning the metal surfaces with a damp cotton swab to remove traces of the extract, a final step was taken to verify whether the surface was free of bacteria. After the cleaning step, 100 μL of distilled water was applied to the metal surfaces with the plant extracts and then pipetted back (pipetting). After the third pipetting, the liquid retrieved from the metal surfaces in the pipette was transferred to tryptic glucose yeast agar plates. After 24 h of incubation at 37 °C, the Petri dishes were examined for any growing colonies.

## 3. Results

### 3.1. Bioactive Metabolites Having a Potential Role in Biocorrosion Inhibition

Table 2 and Table 3 report the characterization of *A. vera* and *O. ficus-indica* extracts, respectively.

Aloin and aloe-emodin, saponins, and phenolic compounds in *A. vera* and flavonoids (e.g., quercetin), tannins, and polyphenols in *O. ficus-indica* can exert a potential role as antibiofilm/antibiocorrosion agents. The concentrations of the determined secondary metabolites in the crude gel (undried material) were comparable to those in the literature, with significantly higher amounts of flavonoids and carotenoids in *O. ficus-indica*.

### 3.2. Minimal Inhibitory Concentration (MIC)

To determine the MIC of the plant extracts, tube samples NC, PC, 1, 2, 3, and 4 (for *O. ficus-indica*) and NC, PC, 5, 6, 7, and 8 (for *A. vera*) were observed and compared (Figure 1a,b). The absence of turbidity in samples 3 and 7 indicated that the MIC for both extracts was 8.3% (*v*/*v*).

### 3.3. Time-Kill Test

The results of the time-kill tests, expressed as the average bacterial cell counts from three experimental series using both spectrophotometric analysis and the spread-plate method, were analyzed as killing curves by plotting the log_10_ CFU/mL versus time (h). Antimicrobial agents can kill (bactericidal activity) or inhibit bacterial growth (Bacteriostatic activity). A reduction of at least 3 log_10_ colony-forming units per milliliter compared to the initial bacterial count indicates that the agent is bactericidal active, while a reduction of less than 3 log_10_ indicates bacteriostatic activity [31,32].

Figure 2a,b reports the killing curves for spectrophotometric analysis and the spread-plate method for the *O. ficus-indica* extract.

The time-kill profiles exhibit a consistent trend, regardless of the methods used, confirming the validity of both analytical approaches. As expected, bacterial growth in PC increased steadily over time. In contrast, the kill kinetic profiles for the sample treated with extract of *O. ficus-indica* with 16.6% (*v*/*v*), 4, displayed a rapid bactericidal activity for the concentration, showing a ≥3 log_10_ reduction in viable cell count relative to the initial value after 18 h of incubation (Figure 2a,b).

The killing curves for both spectrophotometric analysis and the spread-plate method using A. vera extract are reported in Figure 3a,b. The two analytical approaches are still valid, but in this case the kill kinetic profiles of the samples treated with extracts of *A. vera* with 8.3% and 16.6% (*v*/*v*) concentrations, **7** and **8**, displayed a bactericidal activity with a ≥3 log_10_ reduction in viable cell count relative to the initial value after 18 h and 24 h of incubation, respectively.

Comparing the two extracts, we can conclude that *A. vera* exhibits higher bactericidal activity, demonstrating a reduction ≥3 log_10_ even at lower concentrations.

### 3.4. Antibiofilm Performance

#### 3.4.1. Prevention of Biofilm Formation

Figure 4a–c show the differences in bacterial density for samples B1–B4, C1–C4, and Br1–Br4 (see Table 2). Cloudy or opaque areas indicate the presence of *P. aeruginosa* biofilm, while clear or transparent zones suggest the absence of bacterial growth. Dotted or uneven patterns represent areas where the biofilm was partially disrupted or eliminated.

The density is the highest for the positive control samples B1, C1, and Br1, while for samples B2, C2, and Br2, it is practically zero, as the antibiotic kills the bacteria. In the other samples, B3–B4, C3–C4, and Br3–Br4, the density decreased significantly compared to the positive samples. The best results were obtained with the extract of *A. vera* regardless of the type of metal, i.e., samples B4, C4, and Br4.

#### 3.4.2. Elimination of Formed Biofilm

Figure 5a–c shows the results obtained for the stained samples (see Section 2.6) by treating the metal samples with the plant extracts after forming a bacterial biofilm.

Regardless of the type of metal, the number of bacterial cells (blue-stained) is higher in the three positive controls and untreated samples and lower in the treated samples. Also, in this case, the extract of *A. vera* is more effective than that of *O. ficus-indica*.

#### 3.4.3. Surface Alterations and Residue Removal After Treatment

The samples prepared to test the effectiveness of the plant extracts on pre-formed biofilms appeared as shown in Figure 6 after incubation and transfer of the bacterial colonies.

It is important to note that, although the bacterial load on the three treated samples was significantly reduced, complete bacterial elimination was not achieved.

As shown in Figure 6, the surface of all samples was stained: the visible stain on the untreated sample indicated the formation of a biofilm, while the stains on the treated samples suggested that the plant extracts, used as corrosion inhibitors, also altered the surface appearance.

Residual traces of the plant extracts were easily removed by gently wiping the surface with a soft, slightly moistened cotton cloth. After cleaning with a cotton cloth, samples were taken from the metal surfaces with the plant extracts to determine the presence of any bacterial colonies on the surface, and no bacterial colonies were detected on the plates with agar. This gentle mechanical action was sufficient to remove most of the surface residues and stains, restoring the metal samples to an acceptable condition, even for purposes of conservation and restoration.

For safety reasons and to ensure the inactivation of any remaining possible bacteria, all samples, including controls, were sterilized by autoclaving before further handling.

## 4. Discussion

The secondary metabolites present in *A. vera* and *O. ficus-indica* extracts are likely responsible for their antibiofilm and biocorrosion inhibition effects. In *A. vera*, aloin and aloe-emodin disrupt bacterial membranes and inhibit biofilm formation, while saponins interfere with microbial adhesion and biofilm matrix integrity. Phenolic compounds contribute to oxidative stress in bacteria and inhibit quorum sensing [23,24,33]. In *O. ficus-indica*, flavonoids such as quercetin inhibit bacterial growth and biofilm formation, tannins precipitate microbial proteins and disrupt biofilm structure, and polyphenols interfere with microbial metabolism and adhesion [22,34]. These bioactive compounds provide a comprehensive approach to preventing and mitigating biocorrosion on metal surfaces.

The antibacterial and antibiofilm properties of *A. vera* and *O. ficus-indica* extracts against *P. aeruginosa* were confirmed through both the spread-plate method and spectrophotometric analysis. The secondary metabolites present in *A. vera*, such as anthraquinones, phenolic compounds, saponins, and carotenoids, are likely to contribute to its strong antibacterial activity. Similarly, the phenolic compounds, flavonoids, and tannins in *O. ficus-indica* play a crucial role in inhibiting bacterial growth and the formation of biofilms. These findings align with previous studies that have demonstrated the antimicrobial properties of these plants [35,36,37,38,39].

The spread-plate method and spectrophotometric analysis were employed to determine if one method could be advantageous when working with a plant extract. The results show no differences between the two approaches. While the spectrophotometric method offers the advantage of being faster, the spread-plate method requires only basic laboratory equipment, making it accessible and cost-effective. Both methods proved to be consistent and repeatable.

The anti-biofilm tests were presented in the second part of this study. First, we demonstrated that *P. aeruginosa* forms a biofilm on all tested copper metal samples, although these are often described in the literature as metals with antibacterial properties [40,41]. This result confirms our original hypothesis that these metals are susceptible to microbial corrosion despite their use as antimicrobial agents. A relatively recent review on this topic reports that microorganisms can degrade the quality of many metals, including aluminum, copper, nickel, and titanium, despite most microbiological research focusing on the role of microorganisms in the corrosion of iron-containing ferrous metals, such as carbon and stainless steel [42].

Indeed, it is frequently reported in the literature that the formation of a biofilm on the metal surface is the first step of bio-corrosion, implying that the metabolites present at the biofilm–metal interface are likely to be more relevant to the processes related to microbial-induced corrosion [43,44]. Inhibiting the formation of a biofilm or treating an existing biofilm can prevent bio-corrosion and metal deterioration. The *A. vera* and *O. ficus-indica* extracts were tested for these purposes.

The potential of these two plant extracts as metal corrosion inhibitors [9,10,11,12,13,14] is well known, while their possible impact on biocorrosion had not been thoroughly explored until this study. Various methods, including chemical techniques, spectrophotometric methods, electron microscopy techniques, and stereomicroscopy methods, can typically be employed to analyze antibiofilm activities. In our study, the precise quantification of bacterial cells was not the primary objective; instead, it was on comparing treated and untreated samples to assess where the biofilm had decreased.

Our results show that both plant extracts are effective in preventing biofilm and thus biocorrosion when applied before bacterial colonization and that they can effectively eliminate *P. aeruginosa* biofilm once it has already formed on metal surfaces.

After biofilm removal, we examined the metal surfaces for possible future applications in conserving and restoring metal artifacts. Stains were observed on the treated metals, but these could be partially removed with a soft cotton cloth moistened with distilled water. This finding aligns with previous studies that have demonstrated the reversibility of *O. ficus-indica* extract [21].

Due to their low thickness, the protective layers of green corrosion inhibitors of archeological metal artifacts are usually invisible. However, in other cases, the inhibitors lead to visible changes, and all layers remain chemically stable in the environment. Due to their thinness, they are not resistant to mechanical removal [18].

In these cases, standard conservation and restoration cleaning techniques should be applied to copper and copper alloys to ensure complete removal of active corrosion.

The application method is crucial in ensuring uniform coverage and minimal visual alteration. In previous tests conducted by some of the authors (unpublished), the same extracts were applied by brushing or immersing the metal samples in the extract solution for two minutes. The brushed samples exhibited surface discoloration due to uneven application, as the liquid collected in certain areas due to surface tension. In contrast, immersion resulted in a more uniform distribution of the extract. Since conservation practice prioritizes preserving an object’s original appearance, brushing may not be a suitable application method for these extracts [45].

These observations lead us to believe immersion would have provided a more homogeneous application and minimized surface alterations in these tests.

## 5. Conclusions

This study aimed to evaluate the potential of *A. vera* and *O. ficus-indica* extracts as biobased agents for the conservation of copper-based cultural heritage metals, with a specific focus on their antibacterial and antibiofilm activities against *P. aeruginosa*.

The results confirmed that:•*P. aeruginosa* is capable of forming biofilms on copper, bronze, and brass surfaces, contributing to microbial-induced corrosion.•Both *A. vera* and *O. ficus-indica* extracts exhibited significant antibacterial activity, with a minimum inhibitory concentration of 8.3% (*v*/*v*).•*A. vera* demonstrated stronger bactericidal effects than *O. ficus-indica*, achieving ≥3 log_10_ reductions in bacterial counts at lower concentrations and shorter exposure times.•Both extracts effectively prevented biofilm formation and reduced established biofilms on metal surfaces.•The treatments were largely reversible, with minimal surface alteration after gentle cleaning.

These findings support the use of *A. vera* and *O. ficus-indica* as eco-friendly, sustainable alternatives to synthetic biocides in the conservation of metal artifacts. Their dual functionality—as both corrosion and biocorrosion inhibitors—offers a comprehensive approach to preserving cultural heritage metals from both chemical and biological degradation.

Future research should focus on optimizing application methods to minimize visual impact and testing these extracts on actual archaeological objects to assess their long-term effectiveness and reversibility in real-world conservation contexts.

## Figures and Tables

**Figure 1 metabolites-15-00386-f001:**
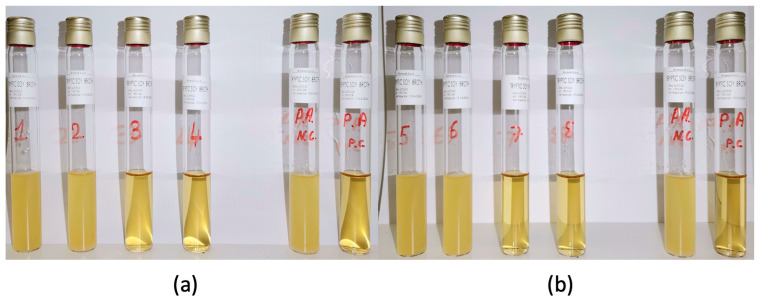
MIC analysis for (**a**) *O. ficus-indica* and (**b**) *A. vera* extracts/P.A: *P. aeruginosa*/1.5: 2.075 (*v*/*v*); 2.6: 4.15 (*v*/*v*); 3.7: 8.3 (*v*/*v*); 4.8: 16.6 (*v*/*v*). NC: negative control—PC: positive control.

**Figure 2 metabolites-15-00386-f002:**
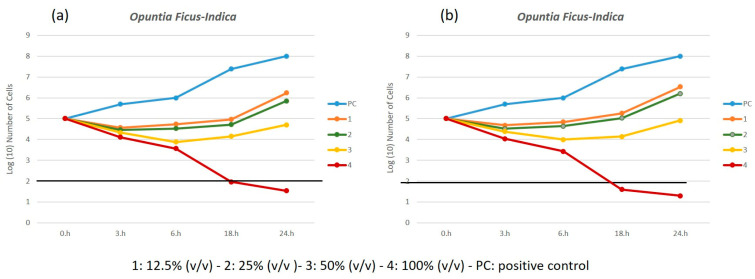
Time kill profiles for samples treated with solutions of *O. ficus-indica.* (**a**) by spectrophotometric analysis and (**b**) by the spread plate method.

**Figure 3 metabolites-15-00386-f003:**
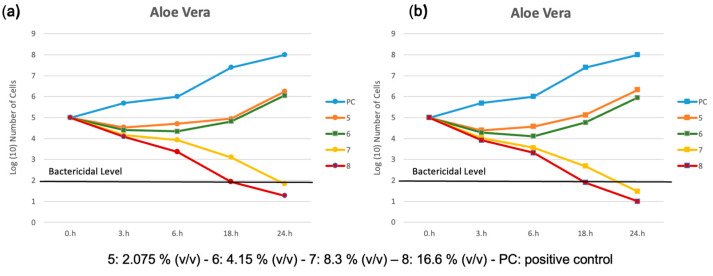
Time kill profiles for samples treated with solutions of *A. vera.* (**a**) by spectrophotometric analysis and (**b**) by the spread plate method.

**Figure 4 metabolites-15-00386-f004:**
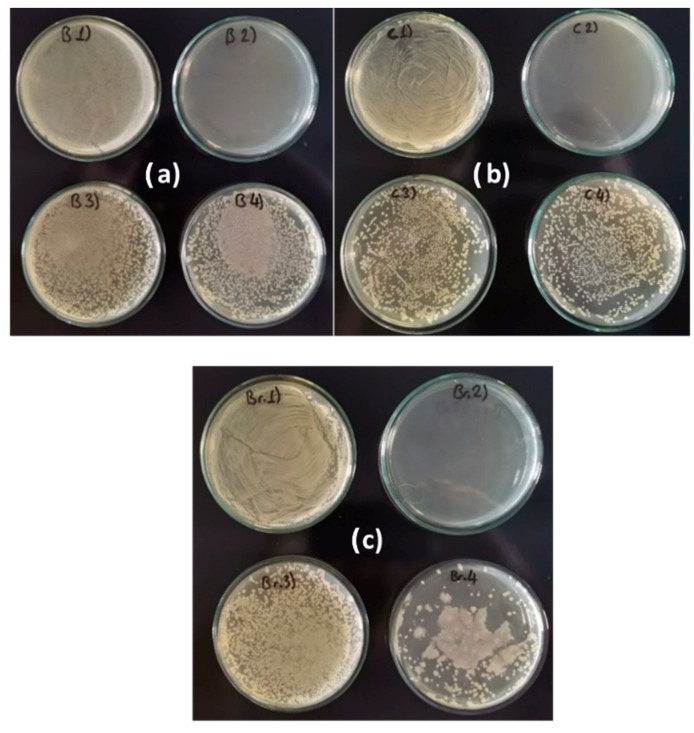
Bacterial densities on (**a**) brass samples, (**b**) copper samples, and (**c**) bronze samples. Composition of treatment solutions: B1, C1, Br 1: Bacteria/B2, C2, Br2: Bacteria + Antibiotic/B3, C3, Br3: Bacteria + *O. ficus-indica* extract/B4, C4, Br4: Bacteria + *A. vera* extract).

**Figure 5 metabolites-15-00386-f005:**
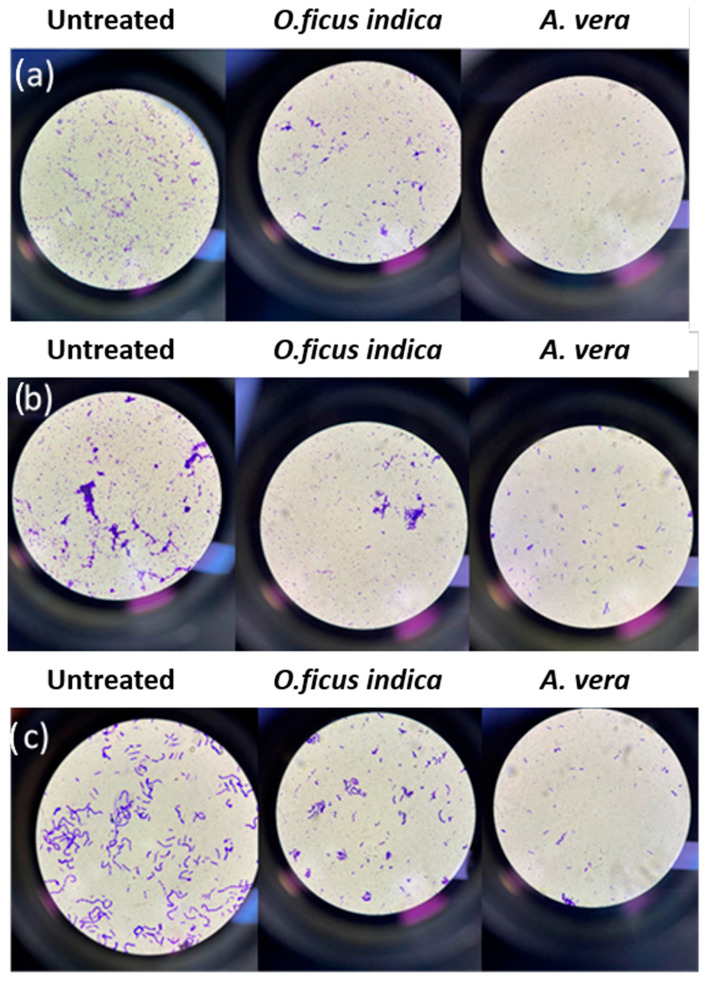
Stained bacterial biofilm from (**a**) copper samples, (**b**) brass samples, and (**c**) bronze samples.

**Figure 6 metabolites-15-00386-f006:**
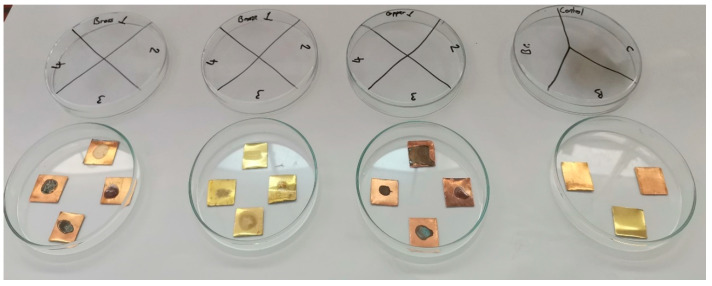
Metal surface condition after biofilm removal and control samples of entreated metals, Bronze (Br), Copper (Cu), and Brass (B). Control samples bacteria: *P. aeruginosa*./1: Bacteria, 2: Bacteria + Antibiotic, 3: Bacteria + *O. ficus-indica* extract, 4: Bacteria + *A. vera* extract.

**Table 1 metabolites-15-00386-t001:** Samples prepared for colony counting.

Metal	Composition of Treatment Solutions			
	* **P. aeruginosa** *	* **P. aeruginosa + Azithromycin** *	* **P. aeruginosa + O. ficus-indica** *	* **P. aeruginosa + A. vera** *
Brass	B1	B2	B3	B4
Copper	C1	C2	C3	C4
Bronze	Br1	Br2	Br3	Br4

**Table 2 metabolites-15-00386-t002:** Characterization of the *A. vera* sample in comparison to existing literature findings.

Compound	Concentration Range (*w*/*w*)	Our Sample (Crude Gel) (*w*/*w*)	Notes
**Aloin**	0.1–0.66% in fresh leaves [23]	0.05%	Anthraquinone with known antibacterial and antifungal activity.
**Aloe-emodin**	0.01–0.1% in fresh leaves [23]	0.02%	Oxidative derivative of aloin; disrupts microbial membranes.
**Phenolic compounds**	0.02–0.2% total phenolics [24]	0.03%	Includes flavonoids and phenolic acids; antioxidant and antimicrobial.
**Saponins**	0.1–0.3% in gel [24]	0.05%	Surface-active agents that disrupt biofilms.
**Carotenoids**	Trace to 0.05% (e.g., β-carotene, zeaxanthin) [24]	0.01%	Antioxidant role: may stabilize metal surfaces.

**Table 3 metabolites-15-00386-t003:** Characterization of the *O. ficus-indica* sample in comparison to existing literature findings.

Compound	Concentration Means	Our Sample (Crude Gel)	Notes
**Total polyphenols**	86.6 mg GAE/100 g FW [22]	74 mg GAE/100 g	Includes hydroxybenzoic and caffeic acids.
**Flavonoids**	13.4 mg QE/100 g FW [22]	24 mg QE/100 g	Includes quercetin and isorhamnetin derivatives.
**Condensed tannins**	8.9 mg TAE/100 g FW [22]	5 mg TAE/100 g	Astringent; can bind to bacterial proteins and enzymes.
**Carotenoids**	0.9 mg β-CE/100 g FW [22]	12 mg β-CE/100 g	Antioxidant: minor role in biofilm inhibition.

GAE = gallic acid equivalent; QE = quercetin equivalent; TAE = tannic acid equivalent; CE = β-carotene equivalent; FW = fresh weight.

## Data Availability

Data supporting the results reported in this article are available at the Department of Art and Restoration, University of Dubrovnik, Croatia, and can be requested to Prof. Lucia Emanuele lucia.emanuele@unidu.hr.

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
