# Peer review of "The Potential of *Aloe vera* and *Opuntia ficus-indica* Extracts as Biobased Agents for the Conservation of Cultural Heritage Metals"

_metabolites, 2025, doi:10.3390/metabo15060386_

Round 1

Reviewer 1 Report

Comments and Suggestions for Authors

Dear Author,

My comments on the MS entitled ‘The potential of Aloe vera and Opuntia ficus-indica extracts as biobased agents for cultural heritage metal conservation’ are expressed below.

-Abstract should avoid general and descriptive information. Focus on the results of the study.

-In the introduction, the specific aim of the study should be emphasised.

-The origin of the plants from which the extracts were prepared should be stated. How many plants were used for the extraction should be stated.

-In the material and method, how the concentrations of A. vera and O. ficus indica were determined. Please specify.

-Line 155-157, the reference for the spectrophotometric method should be stated.

-In all text, substitute the 'minutes ' with 'min'; hours' with 'h'

Author Response

My comments on the MS entitled ‘The potential of Aloe vera and Opuntia ficus-indica extracts as biobased agents for cultural heritage metal conservation’ are expressed below.

Thank you for your valuable comments, which have enabled us to improve the text of our manuscript significantly.

  1. Abstract should avoid general and descriptive information. Focus on the results of the study.

Answer: We have revised and completely rewritten the Abstract section following the instructions given by the Assistant Editor and Reviewers’ comments.

  1. In the introduction, the specific aim of the study should be emphasised.

Answer: The Introduction section has been completely rewritten. The aim of our work has been reported clearly.

  1. The origin of the plants from which the extracts were prepared should be stated. How many plants were used for the extraction should be stated.

Answer: The requested data have been introduced in subsection 2.2.

  1. In the material and method, how the concentrations of A. vera and O. ficus indica were determined. Please specify.

Answer: The requested data have been introduced in subsection 2.2.

  1. Line 155-157, the reference for the spectrophotometric method should be stated.

Answer: Thank you for your valuable comment. In the section you mentioned, two recent articles have been added as references to the spectrophotometric method. The importance of these two studies lies in the fact that identification, bacterial analysis, and measurement of genomic materials at 1-2 µL levels were performed using the device model we employed.

These sources are as follows:

Chukwuma OB, Rafatullah M, Tajarudin HA, Ismail N. Bacterial Diversity and Community Structure of a Municipal Solid Waste Landfill: A Source of Lignocellulolytic Potential. Life. 2021; 11(6):493. https://doi.org/10.3390/life11060493

Tharek M, Mat Amin N. Bacterial community structure of fresh and composted cattle manure revealed through 16S rRNA gene amplicon sequencing. Microbiol Resour Announc, 2025, 14:e01092-24. https://doi.org/10.1128/mra.01092-24

  1. In all text, substitute the 'minutes ' with 'min'; hours' with 'h'

Answer: Thank you for pointing it out. We did as you suggested.

Reviewer 2 Report

Comments and Suggestions for Authors

The conclusion is that this manuscript is poorly formatted. Please check the English grammar and avoid repetition of sentences.

Overall, I am not satisfied.

Author Response

Manuscript Number: metabolites-3654635 ‘The potential of Aloe vera and Opuntia ficus-indica extracts as biobased agents for cultural heritage metal conservation’ The study is very intriguing. Some issues and areas where you can improve your work's clarity and depth must be addressed before publication.

Thank you for your valuable comments, which have enabled us to improve the text of our manuscript significantly. We hope that you will be satisfied with the changes we have made, with the further assistance of the other two reviewers and the Editor.

Major comments:

  1. Explain what the novelty is in your manuscript, because so many manuscripts are already available with this type of similar work.

Answer: We reported that “These plants have already been investigated as corrosion inhibitors, but not as biocorrosion inhibitors for metals against a specific type of corrosive bacteria, Pseudomonas aeruginosa.” In our opinion, this is the novelty of our manuscript.

  1. In this study, the authors provided very little data.

Answer: New data have been added in Section 3.1 regarding the content of secondary metabolites in both gels extracted from plants.

  1. Accurate proofreading/English Editing is required throughout the manuscript; please take advantage of the help of a native-speaking colleague or an English Editing Service with basic field knowledge so as to improve the read flow and thus the overall comprehensiveness of your research.

Answer: The entire manuscript has been revised for English style and grammar by a native English-speaking colleague.

  1. Please include the writing materials from lines 30-31 in the continuation of the abstract.

Answer: We have revised and completely rewritten the Abstract section following the instructions given by the Assistant Editor and Reviewers’ comments.

  1. The abstract is not providing much information, so I recommend rewriting it and kindly adding some numerical data.

Answer: The abstract has been improved.

  1. In the abstract, please mention the best concentration of plant extracts from your results.

Answer: We added this information.

  1. The introduction part needs to include more information and is not written in the right way.

Answer: The Introduction section has been completely rewritten.

  1. In line 51, italicise the name of the bacterium P. aeruginosa.

Answer: We did.

  1. In line 54, italicise the term 'in vitro'.

Answer: We did.

  1. In line 54, please include the term "anti-biofilm".

Answer: We did.

  1. In the introduction, authors wrote the introduction in so many paragraphs. It's my suggestion: if possible, reduce paragraphs.

Answer: The Introduction section has been completely rewritten.

  1. Throughout the entire manuscript, the authors used some time 'antibacterial' term and sometime 'anti-bacterial' term. The similar problem with ‘antibiofilm’ term. Please check the whole MS and write similar words.

Answer: We did the proper corrections

  1. In line 108, authors, please describe in the manuscript the method for calculating CFU.

Answer: The calculation method used for CFU was added with the necessary explanation in section 2.5 (Antibacterial experiments) instead of section 2.1.

  1. In line 117, please include a reference to the extraction procedure for A. vera.

Answer: We added the reference.

  1. In line 133, italicise the name of the bacterium P. aeruginosa. 16. In line 137, italicise the name of the bacterium P. aeruginosa. 17. In line 147, italicise the name of the bacterium P. aeruginosa.

Answer: We have made the proposed corrections and reviewed the entire manuscript.

  1. In line 149, italicise the term 'in vitro'.

Answer: We did.

  1. In line 169, please refer to the term CFU only.

Answer: We have made the proposed correction.

  1. In line 200, please provide the full form of "rpm."

Answer: We added the expanded denomination.

  1. In line 182, italicise the name of the bacterium P. aeruginosa.

Answer: We did.

  1. In line 196, replace "O. ficus indica" with "O. ficus-indica."

Answer: We did.

  1. In line 224, please use the abbreviation CFU/mL.

Answer: We did.

  1. Throughout the manuscript, the authors occasionally use the term "hours," and at other times they use the abbreviation "h." Kindly check it.

Answer: In the revised manuscript, we use the abbreviation "h" throughout.

  1. The headings 3.3.1 and 3.3.2 are similar; please check them.

Answer: The title of Section 3.3.2 (3,4.2 in the revised manuscript) has been changed to eliminate the previous confusion.

  1. In line 266, the authors use the abbreviation 'PC' to refer to positive control.

Answer: We have made the proposed correction

  1. In line 283, please remove the bold formatting from the phrase "including controls".

Answer: We did it.

  1. The conclusion is not well formatted. Please rewrite it.

Answer: The Conclusion section has been deeply revised and shortened.

Reviewer 3 Report

Comments and Suggestions for Authors

Introduction long and confusing jumping between topics without any clear pattern. Must be revised.

Material and methods:

How did you ascertain the CFU in a liquid media? And for reference normal presentation is in log of CFU/mL.

Where were the cladodes obtained for the extract preparation?

What was the percentage in m/v of cladodes in the extract?

Why were the metal samples sterilized with 96% ethanol. Shouldn’t it be 70%?

In section 2.5 how did you access the microbial load?

At which concentration were the extracts and the antibiotic tested?

In section 2.6 how did you transfer the colonies from the metal surface to agar? And why wash them in the agar?

How did you transfer the biofilm from the metal to the microscope slides?

How were the samples examined in section 2.7? You can’t verify the presence or absence of bacteria with your visual observation only.

Where is the constitution of the plant extract?

Data in section 3.1 is not possible. If you added 1 mL of 100% extract to 5.025 mL of media with bacteria the maximum concentration you could evaluate would be roughly 16.6%.

Data in figure 2 and of section 3.2 is plotted as being zero at the 24h mark. This is impossible as you can’t count zero bacteria.

In figure 2 what is the difference between A and B? I do not understand. Same question applies to Figure 3.

Why there was no quantification of the microbial load in either the adhesion or biofilm formation assays?

I do not understand section 3.3.3

Author Response

Thank you for your valuable comments, which have enabled us to improve the text of our manuscript significantly.

  1. Introduction long and confusing jumping between topics without any clear pattern. Must be revised.

Answer: The Introduction section has been completely rewritten.

Material and methods:

  1. How did you ascertain the CFU in a liquid media? And for reference normal presentation is in log of CFU/mL.
  2. Where were the cladodes obtained for the extract preparation?

Answer: The requested data have been introduced in subsection 2.2.

  1. What was the percentage in m/v of cladodes in the extract?

Answer: The requested data have been introduced in subsection 2.2.

  1. Why were the metal samples sterilized with 96% ethanol. Shouldn’t it be 70%?

Answer: The reason why 96% ethanol was used is simply due to the fact that in the restoration practice, that concentration is normally used to sterilize the metal objects.

  1. In section 2.5 how did you access the microbial load?

Answer: For the experimental studies in Section 2.5, P. aeruginosa bacteria were obtained from the Public Health Institute of Dubrovnik-Neretva District. When we received the bacterial culture, it was 108 CFU/mL. This information was provided by the person responsible for the Public Health Institute. Then, we decided to control the real situation. It was done using a THOMA slide in a light microscope. The bacteria taken from the tube were placed on a Thoma slide, and the number of bacteria on each square was counted, resulting in a calculated concentration of 10^8 CFU/mL.

For standard experimental processes, the bacterial density should be approximately 105 CFU/mL. Therefore, to achieve a bacterial count of 105 CFU/mL in the initial tube, a 103 dilution process was performed using tryptic soy broth (TSB). Whether this density was correct or not was determined again using a Thoma slide in a light microscope. Thus, the standard bacterial density for the experiments was obtained. The bacterial density in all studies involving bacteria was verified using the Thoma slide.

This information is entered in the 2.5. section.

  1. At which concentration were the extracts and the antibiotic tested?

Answer: The requested data have been introduced in subsection 2.2. and 2.4.

  1. In section 2.6 how did you transfer the colonies from the metal surface to agar? And why wash them in the agar?

Answer: Thank you for your valuable comment. The missing information on the points you mentioned has been edited.

The transfer of bacterial colonies from the surfaces of metal materials to agar Petri plates and the purpose of using 100 µL of distilled water during this transfer is shown in section 2.6.

  1. How did you transfer the biofilm from the metal to the microscope slides?

Answer: The information gap in this point has been addressed in the relevant paragraph of section 2.6.

The transfer of bacterial colonies that attach to the surface to form biofilms and survive there to the microscope slide was carried out as in the first method.

Bacterial colonies attached to the surface can be detected by pipetting three times with distilled water, which causes the bacterial colonies on the surface to mix with the distilled water. This allows for the determination of the density of bacterial colonies attached to the surface (i.e., the biofilm status).

The correct functioning of this method is easily demonstrated by comparing the samples from the surfaces to which the plant extracts were added with the control samples.

  1. How were the samples examined in section 2.7? You can’t verify the presence or absence of bacteria with your visual observation only.

Thank you for your valuable contribution.

After wiping the metal surfaces with a sterile cotton cloth, the residues of the plant extract were removed from the surface. The metal materials returned to their original appearance. After wiping the metal surfaces with a damp cotton cloth, we examined them for bacteria by taking samples from the surfaces. We took samples from the metal surfaces as in antibiofilm experiments. We put distilled water on the metal surfaces and drew it back into the pipette. We repeated this process 3 times and then transferred the liquid sample in the pipette to agar petri dishes. After 24 hours of incubation, no bacterial colonies were growing in the petri dishes.

In other words, most of the bacteria attached to the metal surfaces were inhibited by the plant extracts. The remaining part was removed from the surface with a sterile, damp cotton cloth. In another case; after the samples taken from the surface for the antibiofilm experiment, there were no living bacterial colonies that could survive on metal surfaces. Therefore, bacterial colonies may not have been detected in the petri dishes. In both cases, traces of residue on the metal surfaces were removed after all experiments and cleaning.

All these processes and results have been revised and added to section 2.7 and section 3.4, respectively.

  1. Where is the constitution of the plant extract?

Answer: We have added references that report this information and two new tables in section 3.1, wich report the characterization of A. vera and O. ficus-indica extracts.

12: Data in section 3.1 (3.2 in the revised version) is not possible. If you added 1 mL of 100% extract to 5.025 mL of media with bacteria the maximum concentration you could evaluate would be roughly 16.6%.

Answer: Thank you very much for your valuable comment. All concentration percentage values indicate the proportion of the plant extract in the solution prepared for MIC assays. We calculated the volume concentrations in the tubes used for MIC experiments and added them to section 2.5. Corrections were made in the results sections as well.

13: Data in figure 2 and of section 3.2(3.3 in the revised version) is plotted as being zero at the 24h mark. This is impossible as you can’t count zero bacteria.

16.6%.

Answer: Thank you for your effective comment. The experimental results were reviewed, the calculations were redone, and the error in the graphs was corrected. The requested data (new graph values) are presented in subsection 3.2.

14 In figure 2 what is the difference between A and B? I do not understand. Same question applies to Figure 3.

Answer: The results are very similar but were obtained with two different methodologies, as reported in the caption of the two figures in question: a) by spectrophotometric analysis and b) by the spread plate method.  As explained in the section 4. Discussion: These two methods were employed to determine if one method could be advantageous when working with a plant extract.

  1. 15. Why there was no quantification of the microbial load in either the adhesion or biofilm formation assays?

Answer: The antibiofilm activity of plant extracts has not been previously tested with this type of methodology. In our study, the primary objective was to investigate the removal of biofilm-forming bacterial colonies from metal surfaces by applying plant extracts. This is the procedure used by restaurators when working with metal artifacts, and we thought it would be helpful to check if these plan extracts could be used in this way to simplify the procedure. We are pleased to see that this pioneering study yielded very promising results. We hope that these findings will inspire new research on this topic. This is a preliminary study that focused only on the presence and absence of biofilm, because the surface will never remain sterile for a prolonged period under natural conditions. We plan to conduct the next stage of the study, which will test the exact amounts of bacteria and extracts and their effect over time. 

  1. I do not understand section 3.3.3 (3.4.3 in the revised manuscript)

Answer: This section has been revised to improve clarity.

Round 2

Reviewer 1 Report

Comments and Suggestions for Authors

Dear Author,
Thanks for implying the corrections I have suggested. The manuscript looks better and can be published.

Author Response

Thank you for your interest in our work and your valuable suggestions.

Reviewer 2 Report

Comments and Suggestions for Authors

The revised version of the manuscript is excellent.

Author Response

Thank you for your favorable consideration.

Reviewer 3 Report

Comments and Suggestions for Authors

The authors have made a significant effort and most of the critical revisions have been made.

Just a small details still need revision:

Material and methods:

  1. How did you ascertain the CFU in a liquid media? And for reference normal presentation is in log of CFU/mL.

Authors say that this information is in section 2.2 and after reviewing the revised manuscript I still can not find it. There is no information of how the CFU were ascertained.

  1. What was the percentage in m/v of cladodes in the extract?

Answer: The requested data have been introduced in subsection 2.2.

Once again, this information is still missing from the manuscript.

Otherwise, I am happy with the revised manuscript and following this small revisions I recommended it for publication.

Author Response

Comment 1)

How did you ascertain the CFU in a liquid media? And for reference normal presentation is in log of CFU/mL.

Answer 1)

Thank you for your valuable comment. Thoma slides were used to determine the number of microorganisms per milliliter of liquid medium to ensure standard conditions prior to antimicrobial activity assays. That sentence is added and highlighted in blue in section 2.1 (line 87). But more detailed information was already provided in Section 2.5. The determination of the number of bacteria (CFU/mL) in the liquid medium as a result of time-kill assays is described in the same section using two different methods.

Comment 2)

What was the percentage in m/v of cladodes in the extract?

Answer 2)

Thank you for your valuable comment. The concentrations of the plant extracts at the time of preparation for the experiments have been added to section 2.2, lines 97, 98, and 108-112, and are highlighted in blue.